# Are Shockley-Read-Hall and ABC models valid for lead halide perovskites?

Alexander Kiligaridis[1], Pavel A. Frantsuzov [2✉], Aymen Yangui [1], Sudipta Seth [1], Jun Li [1], Qingzhi An [3], Yana Vaynzof [3✉] & Ivan G. Scheblykin [1✉]

Metal halide perovskites are an important class of emerging semiconductors. Their charge carrier dynamics is poorly understood due to limited knowledge of defect physics and charge carrier recombination mechanisms. Nevertheless, classical ABC and Shockley-Read-Hall (SRH) models are ubiquitously applied to perovskites without considering their validity. Herein, an advanced technique mapping photoluminescence quantum yield (PLQY) as a function of both the excitation pulse energy and repetition frequency is developed and employed to examine the validity of these models. While ABC and SRH fail to explain the charge dynamics in a broad range of conditions, the addition of Auger recombination and trapping to the SRH model enables a quantitative fitting of PLQY maps and low-power PL decay kinetics, and extracting trap concentrations and efficacies. However, PL kinetics at high power are too fast and cannot be explained. The proposed PLQY mapping technique is ideal for a comprehensive testing of theories and applicable to any semiconductor.

[1] Chemical Physics and NanoLund, Lund University, P.O. Box 118, Lund 22100, Sweden. [2] Voevodsky Institute of Chemical Kinetics and Combustion, Siberian Brunch of the Russian Academy of Science, Institutskaya 3, Novosibirsk 630090, Russia. [3] Integrated Center for Applied Physics and Photonic Materials (IAPP) and Centre for Advancing Electronics Dresden (CFAED), Technical University of Dresden, Dresden, Germany. ✉email: pavel.frantsuzov@gmail.com; yana.vaynzof@tu-dresden.de; ivan.scheblykin@chemphys.lu.se

Semiconducting materials often exhibit complex charge dynamics, which strongly depends on the concentration of charge carriers due to the co-existence of both linear and non-linear charge recombination mechanisms[1,2]. The emergence of novel semiconductors like metal halide perovskites (MHPs), exhibiting intriguing and often unexpected electronic properties[3–10], triggered a renewed interest in revisiting the classical textbook theories of charge recombination and the development of more complete, accurate models[11–17]. Moreover, modern technical advances in experimental and computational capabilities[4,18–21] allow for a detailed quantitative comparison between experiment and theory, far beyond what was once possible.

MHP are a novel solution-processable material class with enormous promise for application in a broad range of optoelectronic devices[22–24]. Driven in particular by their remarkable performance in photovoltaics, with power conversion efficiencies surpassing 25% demonstrated to date[25], significant research efforts have been devoted to study the fundamental electronic properties of these materials[4,5,7,13,15,18,26–30]. It was established that for many MHP compositions—with the most notable example being the methylammonium lead triiodide (MA = $CH_3NH_3^+$, also referred to as $MAPbI_3$ or MAPI)—they can be considered as classical crystalline semiconductors at room temperature, in which photoexcitation leads to the formation of charge carriers that exist independently from each other due to the low exciton binding energy[30]. Consequently, conventional models that describe the charge carrier dynamics are ubiquitously used to describe the dynamics of charge carriers in MHPs[11,13–15,31–39].

Historically, the first model describing the kinetics of charge carrier concentrations in a semiconductor was proposed by Shockley and Read[40] and independently by Hall[41], and is known as the Shockley–Read–Hall (SRH) model. This model considers only the first-order process (trapping of electrons or holes) and the second-order kinetic processes (radiative electron–hole recombination and non-radiative (NR) recombination of the trapped electrons and free holes). It is noteworthy that the SRH model allows the concentrations of free charge carriers to differ due to the presence of trapping. In an intrinsic semiconductor, trapping of, for example, electrons generated by photoexcitation creates an excess of free holes at the valence band. This effect is often referred to as photodoping, in analogy with chemical doping, with the important difference, however, that the material becomes doped only under light irradiation and the degree of doping depends on the light irradiation intensity.

Third-order processes, such as NR Auger recombination, via which two charge carriers recombine in the presence of a third charge that uptakes the released energy, have been recognized as particularly important at a high charge carrier concentration regime. To account for this process, Shen et al., instead of adding the Auger recombination term into the SRH model, proposed a simplified ABC model named after the coefficients A, B and C for the first-order (monomolecular), second-order (bi-molecular) and third-order Auger recombination, respectively[42]. These coefficients are also sometimes referred to as $k_1$, $k_2$ and $k_3$. Importantly, the concentrations of free electrons and holes in the ABC model are assumed to be equal, thus neglecting the possible influences of chemical and photodoping effects. The ABC model is widely applied in a broad range of semiconductors and in particular, is commonly used to rationalize properties and efficiency limits of LEDs[42,43]. The simplicity of the ABC model led to its extreme popularity also for MHPs (see e.g. ref. [15] and references therein) with fewer reports employing SRH or its modifications[13,14,16,17,31,36,39,44,45].

The ABC and SRH kinetic models are typically employed to describe experimentally acquired data such as the excitation power density dependence of photoluminescence (PL) quantum yield (PLQY) measured upon continuous wave (CW) or pulsed excitation, time-resolved PL decay kinetics and kinetics of the transient absorption signal. These models are applied to semi-quantitatively explain the experimental results and extract different rate constants[13–15,20,31–34,46,47], often without necessarily considering the models' limitations. Despite the very large number of published studies describing electronic processes in MHPs using the terminology of classical semiconductor physics, to the best of our knowledge, there have been only very few attempts to fit both PL decay and PLQY dependencies of excitation power using ABC/SRH-based models or at least compare the experimental data with theory[14,16,17,31,39]. These attempts, however, were of limited success because large discrepancies between the experimental results and the theoretical fits were often permitted.

These observations raise fundamental questions concerning the general validity of the SRH and ABC models to MHPs and the existence of a straightforward experimental method to evaluate this validity. To address these concerns, it is necessary to characterize experimentally the PLQY and PL decay dynamics not only across a large range of excitation power densities, but also simultaneously over a large range of the repetition rates of the laser pulses. We note that PL is sensitive not only to the concentrations of free charge carriers but also, indirectly, to the concentration of trapped charge carriers, as the latter influence the former via charge neutrality. Such trapped carriers may also lead to other non-linear processes, for example, between free and trapped charge carriers (Auger trapping[2]), which should also be considered, but are excluded from both the ABC and classical SRH models. To expose and probe these processes, it is most crucial to scan the laser repetition rate frequency in the PLQY measurements, with such measurements, to the best of our knowledge, have not been reported to date.

In this work, we developed a new experimental methodology that maps the external PLQY in two-dimensional space as a function of both the excitation pulse fluence ($P$, in photons/$cm^2$) and excitation pulse frequency ($f$, in Hz). Due to scanning of the excitation pulse frequency over a very broad range, this novel technique allows to unambiguously determine the excitation regime of the sample (single pulse vs. quasi-CW), which is critically important for data interpretation and modelling. Obtaining a two-dimensional PLQY($f,P$) map complemented with PL decays provides a clear and unambiguous criterion to test kinetic models: a model is valid if the entire multi-parameter data set can be fitted with fixed model parameters.

By applying this method to a series of high-quality $MAPbI_3$ thin film samples, which when integrated in photovoltaic devices reach power conversion efficiencies of >20% (Supplementary Note 5), we demonstrate that despite $MAPbI_3$ being extensively studied in numerous publications before, neither ABC nor classical SRH model can fit the acquired PLQY maps across the entire excitation parameter space. To tackle this issue, we develop an enhanced SRH model (in the following, the SRH+ model), which accounts for Auger recombination and Auger trapping processes and demonstrates that SRH+ is able to describe and quantitatively fit the PLQY($f,P$) maps over the entire range of excitation conditions with excellent accuracy. PL decays can be also fitted, albeit, with more moderate accuracy. The application of the SRH+ model allowed us to extract the concentration of dominant traps in high electronic quality $MAPbI_3$ films to be of the order of $10^{15}$ $cm^3$ and to demonstrate that surface treatments can create a different type of trapping states of much higher concentration. Beyond the quantitative success of the extended SRH+ model, we reveal that even this model is not capable to describe the PL decay at high charge carrier concentrations. This means that there must be

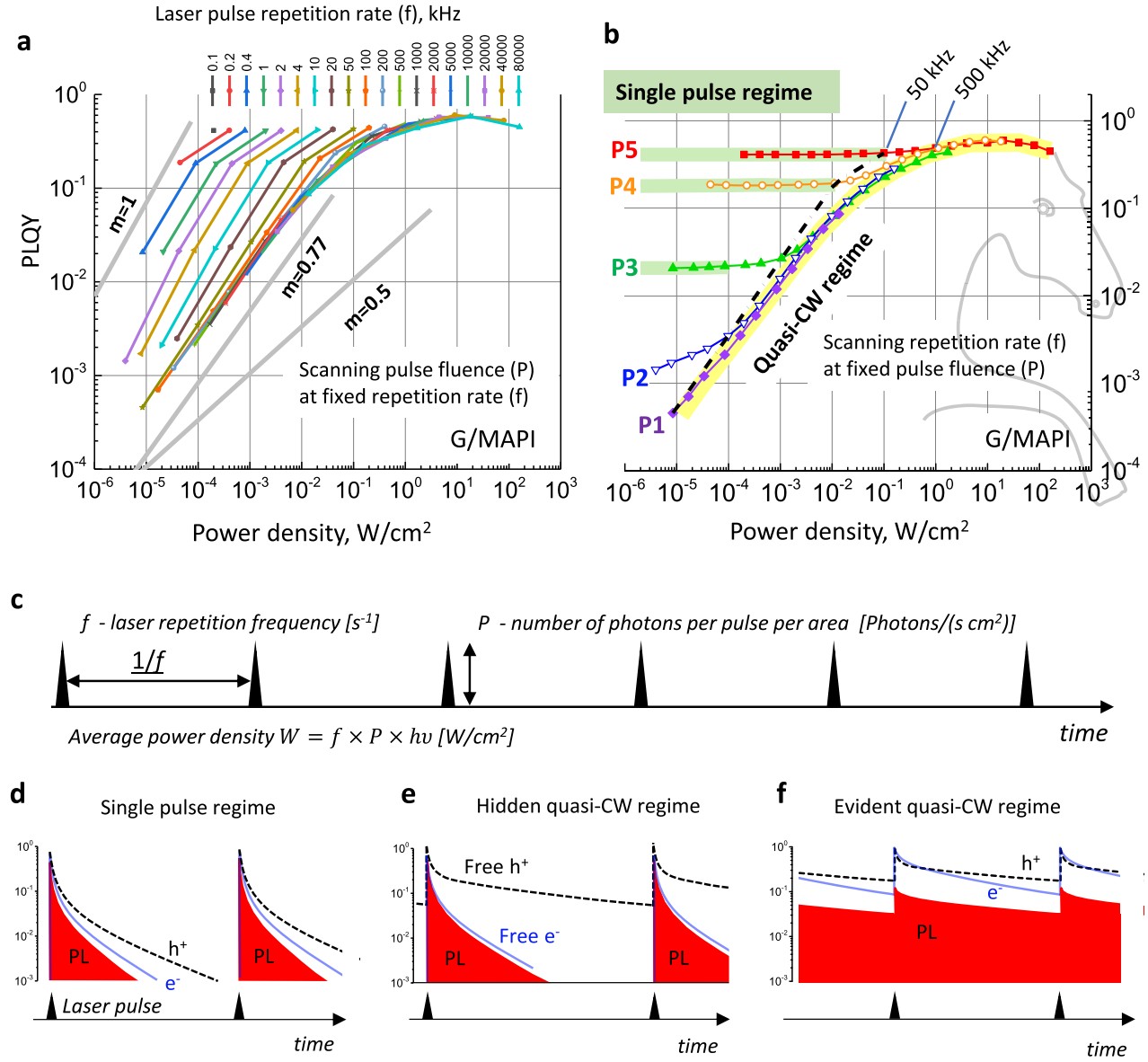

**Fig. 1 PLQY(*f*,*P*) map and illustration of the difference between scanning the pulse repetition rate (*f*) and scanning of the pulse fluence (*P*). a** PLQY (W) dependence plotted in the traditional way (*P*-scanning) for 19 different pulse repetition rates. The datapoints measured at the same frequency are connected by lines, the sample is MAPbI₃ film grown on glass (G/MAPI). The apparent slope of these dependencies (*m*, PLQY ~ W^*m*) depends on the range of *W* and the value of *f* and can be anything from 1 to 0.77 for this particular sample. **b** The same data plotted in the form of a PLQY(*f*,*P*) map where data points measured at the same pulse energy (*P1*, *P2*, ..., *P5*) are connected by lines (*f*-scanning). Data points measured at 50 kHz frequencies are connected by a dashed-dotted line. **c** The excitation scheme. Illustrations of PL decays in the single pulse (**d**) and quasi-CW (**e**, **f**) excitation regimes. Here e⁻ trapping is assumed leading to h⁺ photodoping.

further non-linear mechanisms that influence charge dynamics at high charge carrier concentrations in MAPbI₃. Therefore, further theoretical work is necessary to identify the additional physical process or processes which must be considered in order to completely elucidate the charge dynamics in MAPbI₃.

## Results

**PLQY(*f*,*P*) mapping and elucidation of the excitation regime.** The acquisition of a PLQY(*f*,*P*) map occurs by measuring the intensity of PL as a function of pulse repetition rate (*f*, s⁻¹) for a series of fixed pulse fluences (*P*, photons/cm²). PL intensity is then plotted as a function of the time-averaged excitation power density $W = fPh\nu$ (W/cm²), where $h\nu$ is the excitation photon

energy, see further details in Supplementary Notes 1–3. Figure 1b presents PLQY(*f*,*P*) map for a bare MAPbI₃ film, while Fig. 1a presents the same data in the traditional way as a series of PLQY (W) dependencies for different frequencies. We use 19 frequencies ranging from 100 Hz to 80 MHz, which corresponds to a lag between pulses varying from 10 ms to 12.5 ns. In our experiments after scanning the frequency for a certain value of *P*, it is then changed to the next value and the scanning procedure is repeated. The pulse fluence ranges over four orders of magnitude (*P1* = 4.1 × 10⁸, *P2* = 4.9 × 10⁹, *P3* = 5.1 × 10¹⁰, *P4* = 5.5 × 10¹¹ and *P5* = 4.9 × 10¹² photons/cm²). Such fluences, in the single-pulse excitation regime (see below), correspond to charge carrier densities of 1.04 × 10¹³, 1.24 × 10¹⁴, 1.3 × 10¹⁵, 1.37 × 10¹⁶ and 1.24 × 10¹⁷ cm⁻³, respectively. For clarity, in Fig. 1 and in all the

following figures in the manuscript, the data points measured at the same pulse fluence $P$ are shown by the same colour: $P1$—violet, $P2$—blue, $P3$—green, $P4$—orange, $P5$—red. The family of lines connecting points with $P$ = constant and $f$ scanned make together a pattern that resembles "a horse neck with mane" as illustrated in Fig. 1b.

The acquisition of a PLQY($f,P$) map is fully automated (Supplementary Note 2) and includes precaution measures that minimize the exposure of the sample to light, while controlling for photo-brightening or darkening of the samples (Supplementary Note 4). Such measures ensure that PLQY maps are fully reproducible when re-measured again on the same spot (see Supplementary Fig. 4.2). These precautions were absolutely necessary for obtaining a consistent data set, because light-assisted transformation of defect states due to ion migration may significantly influence the photophysics of perovskite materials[10,48,49] making any theoretical analysis unfeasible. We also note that the high degree of uniformity of our samples leads to very similar PLQY maps being measured on different areas of the sample (see Supplementary Notes 2 and 4).

PLQY($f,P$) map for bare MAPbI$_3$ samples (G/MAPI) is presented in Fig. 1. A traditional representation of these data is shown in Fig. 1a, which displays a series of PLQY(W) curves—each for one of the 19 different repetition rates used in our experiment. Overall, such representation shows only minor differences between the curves, in terms of their slope and curvatures, apart from a noticeable horizontal shift at sufficiently low frequencies. By approximating the PLQY to vary as $W^m$ over a limited power interval, we observe that the slope $m$ varies between 0.77 at high repetition rates to approximately 1 at low repetition rates for this sample. Traditional representations of PLQY(W) plots for the other samples investigated in our study are shown in Supplementary Note 8, where, for example, a PMMA coated MAPbI$_3$ sample shows the slope $m$ ranging from 0.5 to 1 (see Supplementary Fig. 8.1c). Such a traditional representation of the PLQY($f,P$) map does not offer a clear interpretation of the data, making it difficult to elucidate the charge carrier dynamics.

An alternative representation of the PLQY($f,P$) map is shown in Fig. 1b, in which the data points for each laser fluence $P$ are presented as a single curve. Interestingly, the data points for each value of $P$ follow a characteristic line with a specific shape. At low frequencies, and especially at high fluences, the curves are rather horizontal, yet once the frequency $f$ exceeds a certain value, all data points start to follow a certain common dependency, at which the PLQY depends solely on the averaged power density $W = fPh\nu$. The frequency at which this happens depends on $P$, such that, for example, the data obtained at pulse fluence P5 joins at ca. 500 kHz, while the data collected at P1 joins at below 50 kHz (see Fig. 1b).

Critically, such presentation of the PLQY($f,P$) map allows us to immediately distinguish between two principally different excitation regimes for a semiconductor:

1. *Single-pulse regime*: In this regime the repetition rate of the laser is so low, that PLQY values and PL decays do not depend on the lag between consecutive laser pulses. In other words, the excited state population created by one pulse had enough time to decay to such a low level, that it does not influence the decay of the population generated by the next pulse (Fig. 1d). In this case, PLQY does not depend on the lag between pulses (i.e. the pulse frequency). This regime is observed when PLQY follows the horizontal lines upon frequency scanning (highlighted in green in Fig. 1b).

2. *Quasi-continuous wave (quasi-CW) regime*: In this regime, the decay of the population generated by one pulse is dependent on the history of the excitations by previous

pulses. This happens when some essential excited species did not decay completely during the lag time between the laser pulses (Fig. 1e, f). In this regime, the data points follow the same trend and fall on the line highlighted in yellow in Fig. 1b. The transition between the two regimes occurs when the data points at fixed values of $P$ start to match with each other upon increasing the $f$.

Examining the vast literature of MHPs reveals that, to the best of our knowledge, no study has utilized such a broad range of pulse repetition rates $f$ when measuring PLQY(W). Without scanning of $f$ over a significant range of values, a distinction between the single pulse and quasi-CW regimes is not possible, and this reflects the current situation in the literature where the standard scanning over $P$ is implemented with, at best, a few different repetition rates of a pulsed laser, which is sometimes complemented by excitation by a CW laser[16,17,39]. For example, Trimpl et al.[39] studied FA$_{0.95}$Cs$_{0.05}$PbI$_3$ with the focus on temperature-dependent PL decay kinetics measured at three repetition rates (61.5, 250, and 1000 kHz) and PLQY at one repetition rate and three pulse fluences approximately corresponding to P2, P3 and P4 in our experiments. A qualitative similarity between PLQY predicted from PL decay kinetics and experiments data was obtained and temperature dependence of the model parameters was extracted. The condition for charge accumulation (photodoping) in this work was addressed solely using PL decays where an initial fast drop at the ns time scale clearly visible at low temperature was assigned to trapping[39]. In another example, Kudriashova et al. studied PL decay over a quite broad pulse repetition rate range (10 kHz–10 MHz) in order to distinguish between surface and bulk charge recombination in MAPbI$_3$ films with charge transport layers, however, PLQY was not measured in this study[34]. In general, these studies addressed the important question of the excitation regime within the limits of their experimental approaches, however, the only robust way to clarify the excitation condition for a given pulse fluence is to explicitly scan $f$ while detecting PLQY. One may assume that choosing a low repetition rate guarantees that the excitation is in the single-pulse regime. However, this is not true. As Fig. 1a and b show, if the system is at the single-pulse regime at a high pulse fluence at a given repetition rate, there is always such low pulse energy that the excitation regime becomes quasi-CW. The cause for this effect is the presence of a non-exponential decay of the excited state population as will be discussed later. Thus, at a very low repetition rate, the excitation may still be in a quasi-CW regime so long as the pulse fluence is low enough. Without scanning the pulse frequency, this cannot be disentangled. To illustrate this, the data points measured at 50 Hz were connected by a dash-dot line in the PLQY($f,P$) map in Fig. 1b. For a pulse fluence $P1$ (i.e. the lowest fluence), the excitation regime is quasi-CW. Increasing the pulse fluence by an order of magnitude ($P2$) brings the system close to the single-pulse regime, with further increase of the pulse fluence ($P3$, $P4$ and $P5$) making the excitation fall purely in the single-pulse regime. We highlight the existence of a rather extended intermediate region, at which the regime is neither a single pulse nor quasi-CW. For example, for the pulse fluence $P2$ (charge carrier concentration $\approx 10^{14}$ cm$^{-3}$), this intermediate region starts at 50 kHz and continues down to at least 2 kHz. We underscore that in order to identify the excitation regime without any additional assumptions, one must scan the pulse frequency and measure PLQY. As a result, the PLQY($f,P$) mapping technique described here allows for an unambiguous and very easy discernment between the single pulse and quasi-CW excitation regimes.

**PLQY($f,P$) maps and PL decays($f,P$) of polycrystalline MAPbI$_3$.** Figure 2 compares the PLQY($f,P$) maps measured for MAPbI$_3$

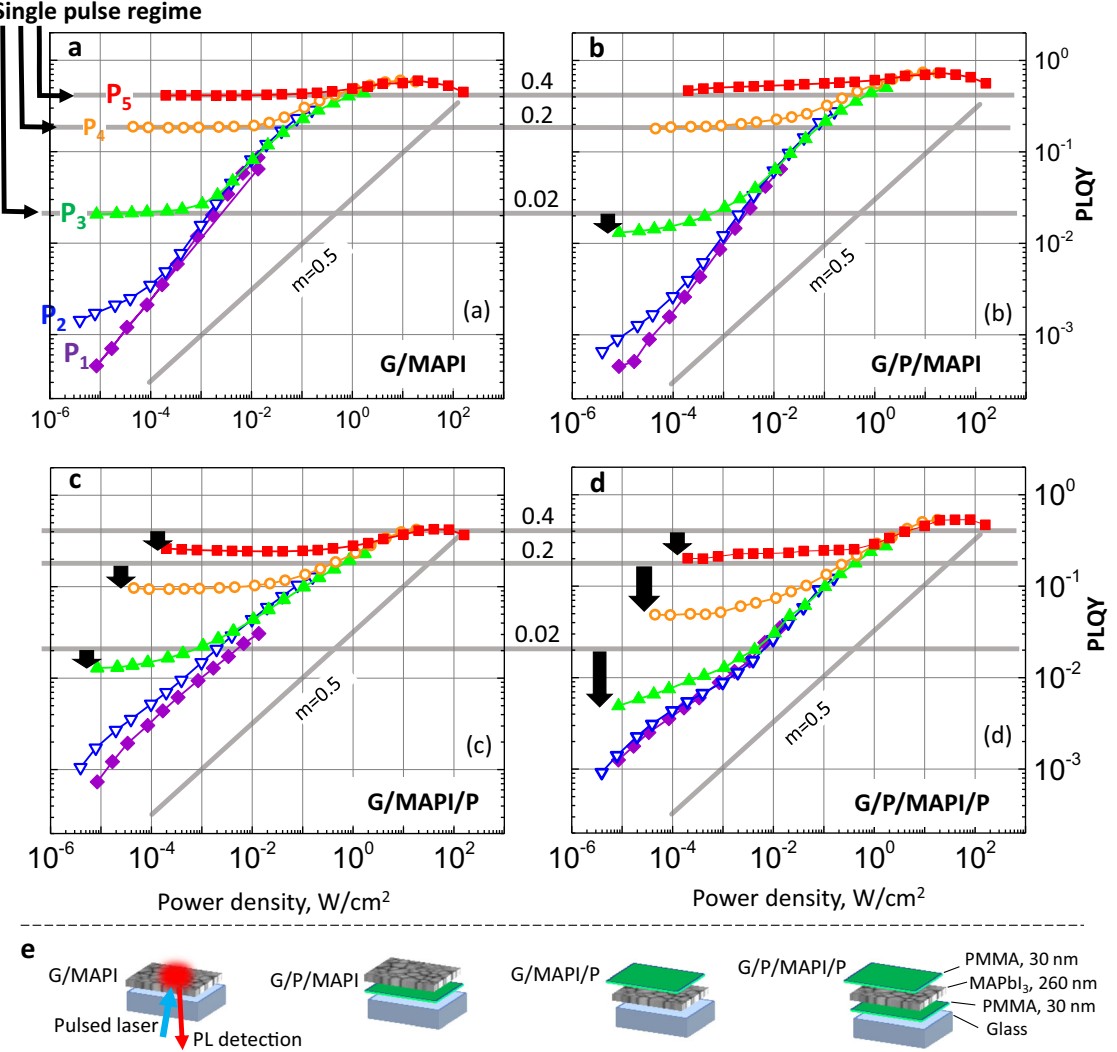

**Fig. 2 PLQY maps of the samples under study plotted in the same scale for comparison. a** Glass/MAPI, **b** glass/PMMA/MAPI, **c** glass/MAPI/PMMA, and **d** glass/PMMA/MAPI/PMMA. The horizontal grey lines show the values of PLQY (0.4, 0.2 and 0.02) in the single-pulse regime for the glass/MAPI sample (**a**) to set the benchmarks. Deviations from these values for other samples are shown by black arrows. The tilted grey line is the $W^{0.5}$ dependence as predicted by the SRH model. It is shown to see better the difference in the quasi-CW regime from sample to sample. The pulse fluence (P1–P5) is indicated by the same colour code (shown in **a**) for all PLQY maps. Panel **e** shows the structure of the samples and geometry of the measurements.

films prepared with four different combinations of the interfaces (Fig. 2e and Supplementary Note 5): MAPbI$_3$ deposited on glass (G/MAPI), MAPbI$_3$ deposited on PMMA-coated glass (G/P/MAPI), MAPbI$_3$ deposited on glass and coated with PMMA (G/MAPI/P) and MAPbI$_3$ deposited on PMMA/glass and then coated by PMMA (G/P/MAPI/P). All samples exhibit the same PL and absorption spectra (Supplementary Note 6). Scanning electron microscopy (SEM) images show that all samples exhibit a very similar microstructure, which is not affected by the presence of PMMA layers (Supplementary Note 6). Despite all these similarities, the PLQY(f,P) maps are clearly different. To emphasize the differences, we added three horizontal lines that mark the PLQY at the single pulse regimes for the pulse fluences P3, P4, and P5 for the G/MAPI sample in Fig. 1a. Black arrows highlight the reduction in PLQY in the single pulse regimes when compared with G/MAPI sample.

The decrease of PLQY upon the addition of PMMA differs for different values of P. Moreover, when comparing the slope m of the quasi-CW region in (a) and (b) with that of (c) and (d), it is evident that it is strongly influenced by the exact sample stack. To visualize this difference, a line with the slope of $m = 0.5$

(i.e. PLQY ~ $W^{0.5}$) is shown in each plot. The PLQY(f,P) map is most affected when MAPbI$_3$ film is coated by PMMA, while its presence at the interface with the glass substrate has only a minor effect.

Similar to the PLQY maps, PL decay kinetics also depends on the pulse fluence and excitation regime (single pulse vs. quasi-CW). Such kinetics should be considered together with PLQY(f,P) map to complete the physical picture of charge recombination. Figure 3 shows PL decays measured at $f = 100$ kHz and pulse fluences P2 (low) and P5 (high). MAPbI$_3$ films deposited on glass (G/MAPI) exhibit the slowest of all PL decay kinetics both at a low and a high pulse fluences. The addition of PMMA to the sample stack accelerates the PL decay with the shortest decays observed for G/P/MAPI/P samples.

The observation that modification of the sample interfaces by PMMA results in a faster PL decay not only for the low, but also for the high (P5) pulse fluence is particularly interesting. While the influence of surface modification on NR recombination at low charge carrier concentrations is expected due to the changes in trapping, the same is not expected to occur at high pulse fluences. It is generally considered that at such fluences, the decays will be

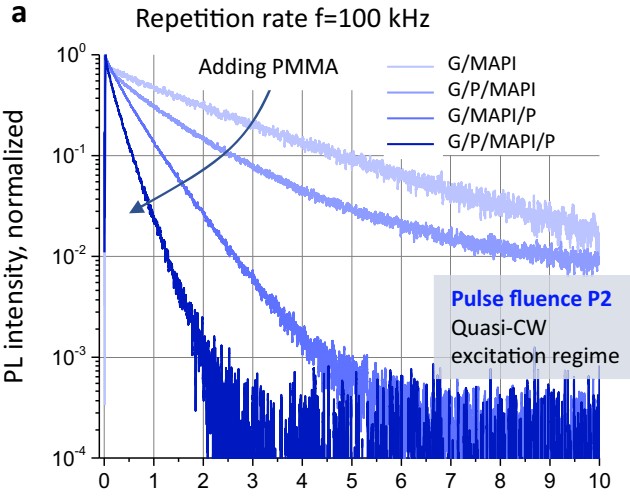

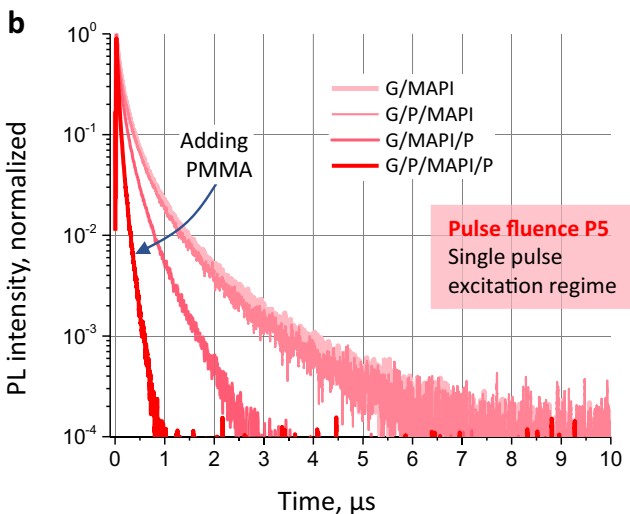

**Fig. 3 PL decays of all samples at 100 kHz repetition rate (10 μs distance between the laser pulses). a** Low pulse fluence (P2). **b** High pulse fluence (P5). Note that all decays in **a** are in the quasi-CW excitation regime, while all decays in **b** are in the single-pulse excitation regime. Adding PMMA accelerates the PL decay.

solely determined by non-linear processes such as Auger recombination and are thus not influenced by surface treatments. However, the change in decay dynamics in PMMA interfaced MAPbI₃ serves as the first indication that additional non-linear processes that involve trap states must be at play.

The second interesting observation is that according to the PLQY(*f*,*P*) map, the repetition frequency 100 kHz used for the PL decay measurements falls in the quasi-CW excitation regime for the low pulse fluence *P2*, but in the single-pulse excitation regime for high pulse fluence P5. It is remarkable, however, that the PL intensity in the quasi-CW regime (Fig. 3a) decays until the next laser pulse by almost two orders of magnitude for MAPbI₃ without PMMA and by four orders of magnitude for the sample coated with PMMA. This is an excellent example for the inability to correctly assign the excitation with *P2* fluence to the quasi-CW excitation regime without the knowledge gained from the PLQY(*f*,*P*) map, considering the population observed in the PL kinetics decays completely prior to the arrival of the next pulse. The cause for the quasi-CW regime, in this case, is the presence of a population of trapped carriers which lives much longer than 10 microseconds and that influences the dynamics via

photodoping[9,13,14]. This example illustrates the 'hidden quasi-CW regime' shown schematically in Fig. 1e (see also Supplementary Note 7). These effects will be quantitatively explained by the theory detailed in the next section.

## Theory and modelling

*Kinetic models: from ABC and SRH to SRH+.* Figure 4a schematically illustrates the key processes included in the ABC, SRH and extended SRH (SRH+) kinetic models. The SRH+ model contains terms for radiative (second-order $k_r np$) and NR (all other terms) recombination of charge carriers. Note, that the processes included in the SRH+ model also naturally include photon re-absorption and recycling as discussed in detail in Supplementary Note 9.1 and Supplementary Note 10 leading to effective renormalization of $k_r$ and $k_E$ rate constants. NR recombination occurs via a trap state or due to Auger recombination. The trapping process can be linear and quadratic (Auger trapping). We note that we consider only one type of band-gap states. It is assumed that these states are placed above the Fermi level (electron traps), but that they are deep enough to make thermally activated de-trapping negligible. Similarly, one could consider hole traps instead under the same conditions—the equations are symmetric in this regard. Auger trapping refers to the process by which the trapping of a photoexcited electron provides excess energy to an adjacent photoexcited hole[2]. The possible importance of this process in perovskites has been suggested in a few publications[46,50]. The complete set of equations and additional description is provided in Supplementary Note 9. We note that in the SRH and SRH+ models, the complete set of equations for free and trapped charged carriers is solved, contrary to the studies where equations for only one of the charge carriers (e.g. electrons) are used (see ref. [44]). The latter simplification can work only if the concentration of holes is very large and constant (for example, in the case of chemical doping) which is not applicable for intrinsic MAPbI₃ and other perovskites, see also below. Due to the inclusion of Auger trapping in the SRH+ model, setting the parameter $k_n$ to infinity reduces it to the ABC model (see Supplementary Note 9.6), where the coefficient *B* contains both radiative and NR contributions. Finally, the SRH+ model reduces to the SRH model by ignoring all Auger processes.

In the considered models the origin of the difference in the concentration of free electrons and holes is the trapping of one of the charge carriers, i.e. photodoping. We do not assume any unintentional chemical doping[45], and this assumption is supported by solid experimental evidence. Indeed, in the case of chemical doping and the presence of electron traps, the PLQY(*W*) in the quasi-CW regime should change from its square root dependence on *W* to either linear (n-doping) or become independent of *W* (p-doping) upon further decreasing of *W* (see Supplementary Note 8). Note also that the situation is symmetrical relative to the type of traps in the material. This behaviour, however, was never observed in our samples where PLQY ∝ $W^m$ at low excitation power with the slope *m* being either 0.5 or 0.77, depending on the sample, without changing upon decreasing of *W* (Figs. 1 and 2). This means, that even if there was unintentional doping in our samples, its level was so low, that we do not observe any of its effects in the PLQY(*f*,*P*) maps (Supplementary Note 9.7).

Photon reabsorption and recycling are considered to be important processes influencing the charge dynamics in MHPs[11,46]. In our experimental study, we compare samples of very similar geometries and microstructure ensuring that the effects of photon reabsorption/recycling remain similar, such that they cannot serve as the reasons for the differences between PLQY(*f*,*P*) maps and PL decay kinetics amongst the different samples. As we discuss in detail

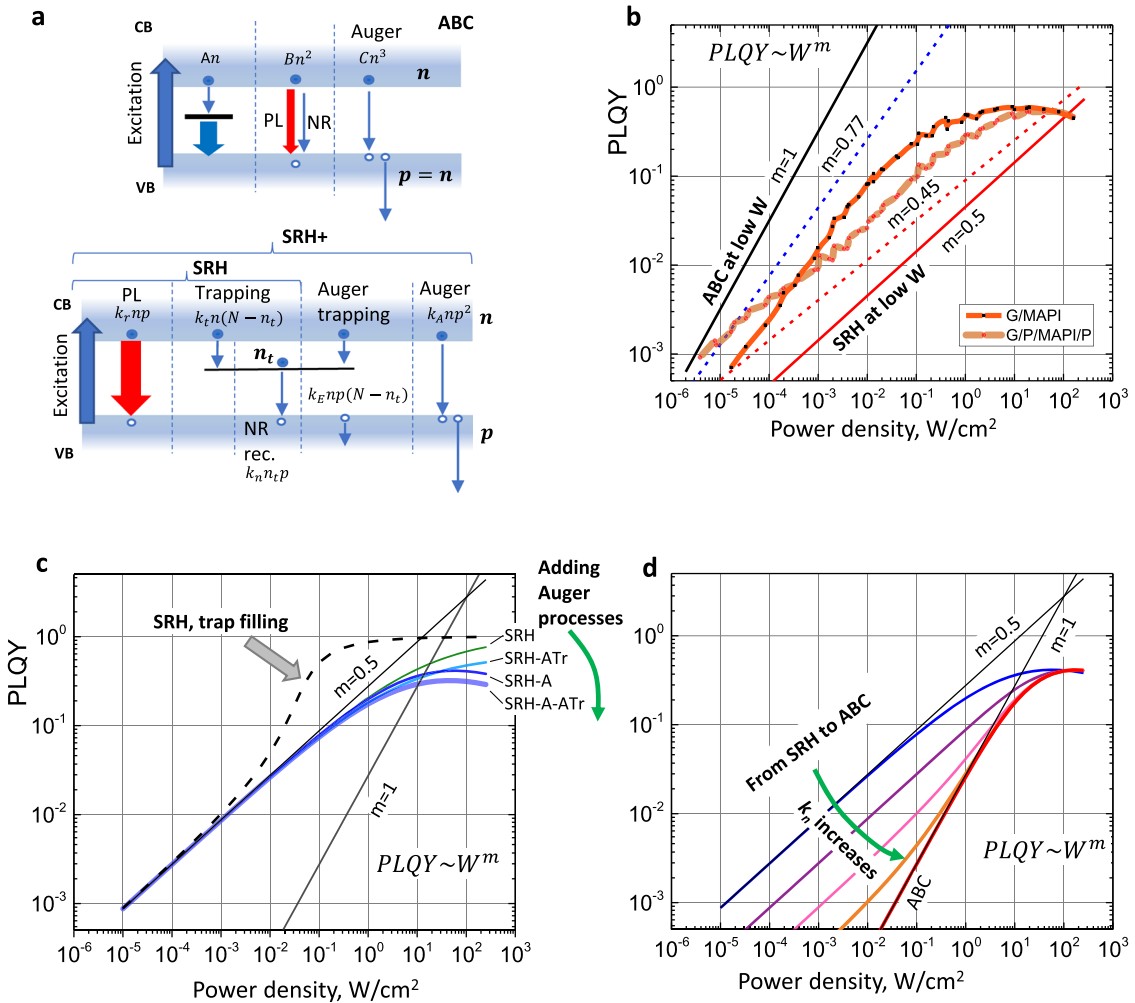

**Fig. 4 CW regimes of the ABC, SRH and SRH+ models and their comparison with the experiment. a** The energy level scheme, the processes and parameters of all models (see the text and Supplementary Note 9 for details). **b** The experimental dependence (G/MAPI and G/P/MAPI/P samples) of PLQY on the excitation power density $W$ in the quasi-CW excitation regime, $m$ is the exponent in the dependence $W^m$. **c** PLQY($W$) in the CW regime for different models and trap feeling conditions. "-A"— adding Auger recombination, "-ATr"—adding Auger trapping (Supplementary Note 11). **d** Evolution of the PLQY($W$) upon the transformation of the SRH model with Auger recombination to the ABC model with increasing of the parameter $k_n$ (see Supplementary Note 11 for the model parameters).

in Supplementary Note 10, effects on the charge dynamics related to photon recycling in broad terms (both far-field (photon reabsorption in the perovskite) and near field (energy transfer) effects), are included in our models via renormalized radiative rate $k_r$ and the Auger trapping coefficient $k_E$, respectively. We also do not explicitly include charge diffusion in the model. The rationale here is that charge carrier diffusion in MAPbI$_3$ occurs so fast that equilibrated homogeneous distribution of charge carriers over the thickness of the film can be assumed at a time scale of 10 ns and longer (Supplementary Note 9.1).

*Applying the ABC, SRH and SRH+ models to the quasi-CW excitation regime.* We first consider the CW excitation regime at low power densities. In this regime, the SRH and SRH+ models are identical since the contribution of Auger processes is largely negligible. Figure 4b shows the experimental dependencies of PLQY on the power density ($W$) for G/MAPI and G/P/MAPI/P samples and Fig. 4c and d show the dependence calculated based on the three different models.

At low power densities, the concentration $n$ is small and PLQY is low. In the ABC model, the main contribution to the recombination rate comes from the first-order term, which is

equal to the photogeneration rate. Thus, $An \propto W$ and consequently $n \propto W$. Therefore, we can write:

$$\text{PLQY} = \frac{\text{flux of emitted photons}}{\text{flux of absorbed photons}} = \frac{k_r n^2}{Bn^2 + An} \approx \frac{k_r n^2}{An} = \frac{k_r n}{A} \propto W$$

In the SRH model, at a very low excitation power density the fastest process is that of trapping of electrons. With most of the electrons trapped and $n_t \approx p$, the trapping rate is equal to the photogeneration rate $k_t nN \propto W$, and the remaining electron density $n \propto W$. The limiting step in the charge carrier kinetics is the NR recombination of the trapped electrons and holes. The rate of this process is equal to the generation rate, therefore $k_n n_t p = k_n p^2 \propto W$, and $p \propto \sqrt{W}$. Thus

$$\text{PLQY} = \frac{\text{flux of emitted photons}}{\text{flux of absorbed photons}} = \frac{k_r np}{k_r np + k_n n_t p}$$

$$\approx \frac{k_r np}{k_n n_t p} = \frac{k_r n}{k_n n_t} \approx \frac{k_r n}{k_n p} \propto \sqrt{W}$$

We refer the reader to the Supplementary Note 9.6 for the detailed derivation of these equations and their applicability conditions. To summarize, at low power densities when PLQY $\ll 1$,

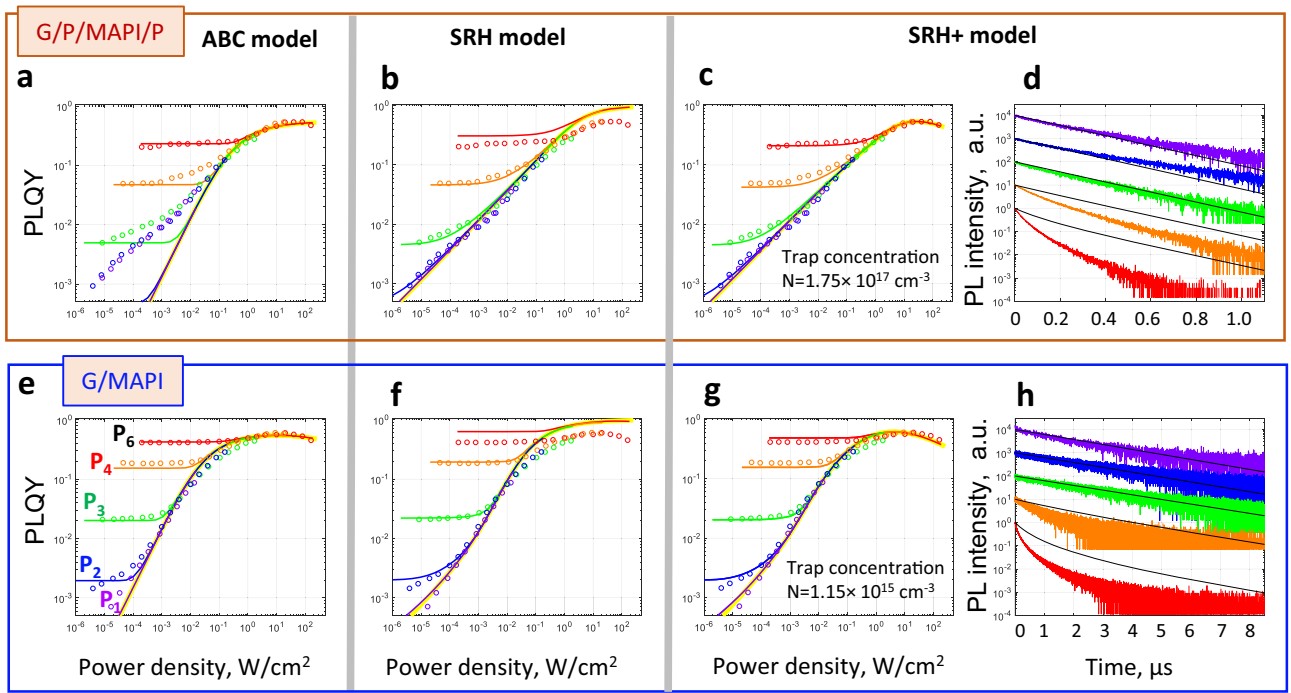

**Fig. 5 Fitting of the PLQY(f,P) maps and PL decays by all models. a** ABC, **b** SRH, **c** SRH+ models applied to the MAPbI₃ film and **e** ABC, **f** SRH, **g** SRH+ models applied to the MAPbI₃ film with PMMA interfaces. In PLQY maps the symbols are experimental points, the lines of the same colour are the theoretical curves. **d** and **h** show experimental and theoretical (black lines) PL decays according to the SRH+ model for both samples, laser repetition rate —100 kHz. The pulse fluences are indicated according to the colour scheme shown in **e** in the whole figure. Theoretical CW regime is shown by the yellow lines in all PLQY maps. The model parameters can be found in Supplementary Note 13.

PLQY($W$) is a straight line in the double logarithmic scale (PLQY $\propto W^m$) with the slope $m = 0.5$ for the SRH and SRH+ models with no trap filling effect (see below) and $m = 1$ for the ABC model[1].

Experimentally, we observe $m \approx 0.45$ for those perovskite samples which are coated with PMMA (e.g. G/MAPI/P is shown in Fig. 4b). This value is in a good agreement to the $m = 0.5$ predicted by the SRH/SRH+ models in the case of the absence of trap filling. However, the other two samples, in which the MAPbI₃ surface is not coated with PMMA, exhibit $m \approx 0.77$ (e.g. G/MAPI sample is shown in Fig. 4b), which lies between the values of 1 and 0.5 predicted by the ABC and SRH/SRH+ models, respectively. These slopes are observed over at least four orders of magnitude in the excitation power density. Based on these results, we must conclude that MAPbI₃ samples with and without PMMA coating behave very differently in the quasi-CW regime.

In the framework of the SRH/SRH+ models, there are two possibilities that would lead to an increase in the coefficient $m$: (i) transformation toward the ABC model and (ii) trap filling effect in the SRH model. Figure 4d shows the transformation of the SRH model, which includes Auger recombination to the ABC model by increasing the parameter $k_n$. At the condition $k_n \gg k_r, k_t$ there is a limited range of excitation power where one can obtain an intermediate slope $m$ laying between 0.5 and 1 for a limited range of $W$ (Supplementary Notes 9.6 and 11).

The second possibility is to allow for the trap filling effect to occur at the excitation power densities which are below the saturation of the PLQY due to the radiative recombination and Auger processes. The effect of trap filling is caused when the number of available traps starts to decrease with increasing $W$. Consequently, the PLQY increases not only because the radiative process becomes faster (quadratic term), but also because the NR recombination (trapping and further recombination) becomes smaller. As the result, PLQY grows faster than $W^{0.5}$ over a certain

range of $W$. The effect is not trivial, because it is not the concentration of traps N as one would think, but rather the relation of $k_t$ to $k_r$ and $k_n$ (the necessary condition is $k_t \gg k_r, k_n$), which determines if the trap-filling effect is observed in PLQY maps or not (Supplementary Notes 9.6, 9.8 and 11).

The trap filling effect is illustrated in Fig. 4c, in which the parameter $k_t$ is increased whilst maintaining all other parameters fixed. Obviously, the resulting dependence is too strong and occurs over a too narrow range of excitation power densities (one order of magnitude) to fit the experimental data directly. Nevertheless, as will be shown below, such processes are present in MAPbI₃ samples which are not coated with PMMA, where PLQY($W$) in the quasi-CW regime deviates from the straight line bending upwards before reaching saturation at high power.

At high excitation densities, non-linear recombination processes begin to be particularly important. Since Auger processes are NR, with further increase of $W$ the PLQY cannot reach unity and instead decreases after reaching a certain maximum. SRH cannot account for this effect considering it does not include any NR non-linear terms and leads to PLQY = 1 at high $W$. The ABC and SRH+ models can potentially describe this regime since they contain Auger recombination terms (Fig. 4c and d).

*Fitting of the PLQY(f,P) maps and PL decays kinetics by ABC, SRH and SRH+ models.* To examine the validity of the three theories, we attempt to fit the experimental PLQY(f,P) plots and PL decays using all models and the results are shown in Fig. 5. Before we discuss the fitting results, it is important to stress that each simulated value of PLQY(f,P) at the PLQY maps and each PL decay curve shown in Fig. 5 are obtained from a periodic solution of the kinetic equations of the corresponding model under pulsed excitation with the required pulse fluence $P$ and repetition frequency $f$. In practice it means that we excited the system again and again until the solution PL($t$) stabilizes and begins to repeat

**Table 1 Comparison of the ability of the three models to describe the PLQY($f$,$P$) maps and PL decays.**

| Observables/regimes | Low and medium excitation pulse fluence power density ($W$ < 0.1 Sun) | High excitation power density (1–300 Sun), high pulse fluence |
|---|---|---|
| **ABC model** | | |
| PLQY($f$,$P$) | Quasi-CW regime—poor or very poor fit strongly depending on the sample. Good fit in the single pulse regime | Very good fit in all regimes |
| PL decays for given PLQY($f$,$P$) | **Cannot predict the PL decays** | **Cannot predict the PL decays** |
| **SRH model** | | |
| PLQY($W$,$f$) | Very good fit in all regimes | **Discrepancy due to exclusion of high order processes** |
| PL decays for given PLQY($f$,$P$) | Very good match | **Underestimation of the initial decay** |
| **SRH+ model** | | |
| PLQY($f$,$P$) | Very good fit in both the quasi-CW and single pulse excitation regimes | Very good fit in both the quasi-CW and single pulse excitation regimes |
| PL decays for given PLQY($f$,$P$) | Very good match | **Underestimation of the initial decay** |

Bold font emphasizes serious deficiencies of the fitting.

itself after each pulse. Details of the simulations are provided in Supplementary Note 12.

When fitting experimental data, it is important to minimize the number of fitting parameters and maximize the number of parameters explicitly calculated from the experimental data. We exploit the experimental data to extract several parameters. First, considering that in all three models, the decay of PL at low pulse fluences is determined exclusively by linear trapping and is thus mono-exponential, we can extract the parameter $k_tN$ of the SRH and SRH+ models. Indeed, such behaviour is observed experimentally for the studied samples (see Fig. 3a) allowing us to use the decays at low pulse energies (P1–P3) to directly determine the trapping rates $k_tN$. We note, however, to obtain the best fit using the ABC model, the PL decays were not used to fix the parameter A. Secondly, in a single pulse excitation regime (i.e. the horizontal lines in the PLQY map), the magnitudes of PLQY at pulse fluence $P3$ and $P4$ allow to determine the ratio $\frac{k_t}{k_tN}$ in the SRH/SRH+ models and the ratio $\frac{k_t}{A}$ for the ABC model. Detailed block schemes of the fitting procedures are provided in Supplementary Note 12.

As has been discussed above, MAPbI$_3$ samples coated with PMMA cannot be described using the ABC model due to mismatch of the slope within the quasi-CW regime (Fig. 5a), while both SRH and SRH+ models are well-suitable in this case (Fig. 5b, c). However, at a high excitation regime (i.e. the saturated region of the quasi-CW and the single pulse regime at $P5$ pulse fluence) SRH+ works much better, highlighting the limitations of the SRH model on its own. Consequently, the entire PLQY($f$,$P$) map of the PMMA-coated films can be fitted using the SRH+ model with excellent agreement between the theoretical and experimental data (Fig. 5c).

The behaviour of MAPbI$_3$ samples whose surface is left bare (where the PLQY($W$) dependence in quasi-CW shows extra up-bending before reaching saturation) can be approximated using the ABC model (Fig. 5e) and well-fitted by the SRH+ (Fig. 5g) model. ABC indeed works quite well with, however, an obvious discrepancy in the tilt of the quasi-CW regime. Very good fit can be obtained by the SRH/SRH+ models by adjusting of the $k_t$, $k_n$ and $N$ to allow for the trap filling effect to occur in the medium excitation power range and, at the same time, making the dynamics closer to that in the ABC model by a relative increase of the recombination coefficient $k_n$ (see the section above and Fig. 5g).

As was mentioned above, the PL decay rate at low power densities (P1–P3) was used to extract the product $k_tN$. This is the only occasion for which the PL decays are used in the fitting procedure of the SRH and SRH+ models. In the fitting procedure for the ABC model the PL decays are not used at all. Upon determining the fit parameters for each of the models, it is possible to calculate the PL decays at each condition and compare them with those decays measured experimentally. Importantly, PL decay rates calculated using the ABC model significantly underestimate the measured decay dynamics at all fluencies (Supplementary Note 13). On the other hand, as is shown in Fig. 5d and h, the SRH+ model (as well as SRH, Supplementary Note 13) fit well the low fluence decay dynamics, but systematically underestimate the decay rate at high power fluences. It is noteworthy that the mismatch of the initial decay rate at the highest pulse fluence reaches a factor ranging from three to five depending on the sample, still significantly outperforming the fit using the ABC model. Insights regarding the applicability of the ABC, SRH and SRH+ models to the PLQY maps and PL decays are summarized in Table 1, see also Supplementary Fig. 13.3.

## Discussion

Scanning the excitation pulse repetition rate as proposed herein represents a novel experimental approach that transforms routine power-dependent PLQY measurements to a universal methodology for elucidating charge carrier dynamics in semiconductors. Adding the second dimension of pulse repetition rate to the standard PLQY($W$) experiment is not just an update, it is a principle, qualitative change of the information content of the experiment. The difference between PLQY($f$,$P$) mapping and the standard PLQY($W$) experiment in the CW regime or at some fixed pulse repetition rate is analogous to the difference between the standard NMR spectrum and 2D NMR spectrum. In our method, we monitor not only the concentrations of free charge carriers, but also the concentration of trapped charges due to the total electro-neutrality of the system. Therefore, together with the time-resolved PL decays, the PLQY map in the repetition frequency—pulse fluence 2D parameter space comprise an experimental series which contains all the information concerning the charge dynamics in a given sample.

We stress the absolute necessity of the unambiguous determination of the excitation regime of the experiment, which would

not be possible without scanning the pulse repetition rate. For example, PL intensity decay kinetic showing the signal decay by several orders of magnitude prior to the arrival of the next laser pulse (Fig. 3a) can still be in the quasi-CW regime due to presence of long-lived trapped charges ("dark" charges). Such trapped charges cause the so-called photodoping effect, which lingers until the millisecond timescale, and thus holds the "memory" of the previous laser pulse, leading to a stark influence on the PLQY map. While the importance of distinguishing between the single pulse and quasi-CW regimes has been noted in several publications before[9,31], it has never been accomplished for MHPs experimentally. Indeed, in none of the published works presenting theoretical fits of experimental PLQY($W$) dependencies was this determination possible simply because either only CW excitation[14,44] or pulse excitation with only one[17,31,39] or two (20 MHz and 250 kHz)[16] repetition rates of the laser pulses were employed.

Understanding the excitation conditions is also critically important for interpretation of the classical experiments in which the PL intensity (or PLQY) is measured as a function of excitation power density ($W$) using a CW light source or a pulsed laser with a fixed repetition rate. Traditionally the intensity of PL is approximated using a $W^{m+1}$ dependence or in case PLQY is measured, with $W^m$ (because PLQY $\propto$ PL/$W$), with both leading to a straight line in the double logarithmic scale[1,13,31,33,47]. According to the SRH and ABC models, approximations like these can be valid for a large range of $W$ at low excitation power density only, when there is no trap-filling effect, Auger processes can be neglected and PLQY is far from saturation. In all other cases, the dependence is not linear in the double logarithmic scale. As discussed above, SRH predicts $m = 0.5$ in the CW excitation regime while ABC always predicts $m = 1$. However, our experiments reveal that when the excitation is pulsed, one can obtain intermediate $m$ values because upon increasing the power density, the experimental excitation regime is almost certainly switched from a quasi-CW to a single pulse. The change of the slope can be seen in Fig. 1a and in Supplementary Note 8. Consequently, the extracted $m$ cannot be reliably used for interpretation of the photophysics of the sample since any value of $m$ can be obtained depending on the conditions of the pulsed excitation.

The main message of our work is that any model of charge carrier dynamics which is considered to be correct should be able to fit not only standard one-dimensional PLQY($W$) data, but also the full PLQY($f,P$) map and PL decays at different powers and pulse repetition rates. This criterion is strict and universally applied. With standard one-dimensional PLQY($W$) data—even upon the inclusion of the PL decay data—one can find several principally different models that are capable of fitting the data. However, when the multi-dimensional data space consisted of the PLQY($f,P$) map and PL decays is available, this ambiguity becomes highly unlikely.

As we have shown above, neither the standard ABC nor the SRH model are capable of describing the complete PLQY maps and predicting PL decays of the investigated MAPbI$_3$ samples. On the other hand, the addition of Auger recombination and Auger trapping processes to the SRH model (SRH+ model) leads to an excellent fit of PLQY maps of all the studied samples. We emphasize that the ($f,P$) space used in this work is very large with $f$ varying from 100 Hz to 80 MHz (6 orders of magnitude) and pulse fluence $P$ changing over 4 orders of magnitude corresponding to charge carrier densities in the single-pulse excitation regime from ca. $10^{13}$ to $10^{17}$ cm$^{-3}$. SRH+ model also agrees well with the PL decay kinetics for low and medium pulse energies (charge carrier concentrations from $10^{13}$ to $10^{15}$ cm$^{-3}$). However, for high pulse fluences ($10^{16}$–$10^{17}$ cm$^{-3}$) the model

underestimates the initial decay rate by up to a factor of five for the higher pulse energies. The Auger rates obtained from the fittings ($2.8 \times 10^{-29}$ cm$^6$ s$^{-1}$ for the PMMA-coated MAPbI$_3$ sample and $1.7 \times 10^{-29}$ cm$^6$ s$^{-1}$ for the bare MAPbI$_3$) are in a reasonable agreement with theoretical estimation $7.3 \times 10^{-29}$ cm$^6$ s$^{-1}$ for MAPbI$_3$ from ref. [51] which is at the low limit from $2 \times 10^{-29}$ to $1 \times 10^{-27}$ cm$^6$ s$^{-1}$ range reported in literature[52]. Note that increasing of the Auger rate constant cannot help because a fit of PL decay will result in lower PLQY than experimentally observed. Therefore, we must conclude that the SRH+ model has limitations.

One possible explanation for the mismatch of decay rates at high excitation powers might be provided by considering experimental errors. It is well documented that the PL of perovskite samples is sensitive to both illumination and environmental conditions, which, may lead to both photodarkening or photobrightening of the sample[9,48,49,53]. To account for these effects, we paid a special attention to monitoring the evolution of the sample under light irradiation throughout the entire measurement sequence. As is shown in Supplementary Note 4, the maximum change in PL intensity during the entire measurement series is smaller than a factor of two. Taking this uncertainty together with other errors inherent to absolute PLQY and excitation power density measurements, missing the decay rates by several times at the highest pulse fluence is not impossible. However, there is strong indication that the discrepancy reflects a problem of the model rather than in the experiment: the deviation between the theoretical and experimental PL decays is systematic. Experimental PL decay rates at high charge carrier concentrations are faster than predicted for all samples despite of the excellent matching of the PLQY($f,P$) maps.

In our future work we are going to test several additional concepts which might help to increase the PL decay rate without a strong effect on PLQY. One of them is based on the idea that at high charge carrier density, the time (few ns) required to reach an equilibration of the charge carrier concentration over the thickness of the sample (300 nm) becomes comparable with the initial fast PL decay induced by Auger. In other words, the diffusion length becomes smaller in the high excitation regime[54]. In this case, diffusion cannot be ignored and an additional PL decay should appear reflecting the decreasing charge carrier concentration due to their diffusion from the initially excited layer determined by the excitation light penetration depth (100 nm) towards the opposite surface of the 300 nm-thick film. This process is often discussed in the context of charge carrier dynamics in large single crystals regardless of the excitation conditions[12]. Supporting this notion is the fact that in order to model a MAPbI$_3$ solar cell under operation[55], a much lower charge carrier mobility (around $10^{-2}$ cm$^2$ V s$^{-1}$) than that obtained spectroscopically (1–30 cm$^2$ V s$^{-1}$)[15] has to be assumed, which suggests that the actual diffusion coefficient might be smaller than expected.

Another possible contributing factor originates from a local charge carrier distribution inside the sample, caused by, for example, funnelling of charge carries due to the energy landscape or/and variations of charge mobilities[13]. Presence of a small fraction of charge carriers concentrated in local nano-scale regions can lead to an apparent fast PL decay at early times, accompanied by a relatively small effect on the total PLQY. In addition, high charge concentrations may cause carrier degeneracy effects. This happens because charge carriers occupy all the possible states with energies below kT (degenerated Fermi gas). Considering that the effective density of states in perovskite materials is relatively low[56], such degeneracy effects should be seriously examined. If present, all rate constants would depend on the charge concentration, which may lead to unexpected effects.

Further investigations will reveal which of these—or other—mechanisms can help in describing of the PLQY(f,P) and PL decay data space.

Despite of the moderate success at high charge concentration regime, the results of the SRH+ fitting still significantly outperform all previous attempts to explain charge carrier dynamics in MAPbI$_3$ samples and allow us to gain valuable insights concerning the photophysics of the samples investigated herein and the roles of traps within them. This is supported by the fact that the effect of charge trapping is the most crucial in the low and middle power ranges where the SRH+ model works very well for both the PLQY maps and PL decays. Note that with the current experimental accuracy we have no reason to complicate the SRH+ model by adding another type of traps and/or thermal de-trapping.

The analysis of PLQY maps reveals that the concentration of dominant traps in high-quality MAPbI$_3$ films (without PMMA coating) is ~$1.2 \times 10^{15}$ cm$^{-3}$. Very recently, practically the same value for trap concentration was obtained using impedance spectroscopy and deep-level transient spectroscopy for MAPbI$_3$ samples prepared by exactly the same method[57]. This concentration is also in excellent agreement with the range of values previously proposed by Stranks et al.[14], where the trap concentration was estimated by assuming that PL decays become non-exponential exclusively due to the trap filling. We note, however, that trap filling is not a necessary condition to observe non-exponentiality in a PL decay. For that to occur, the non-linear recombination rate (radiative, Auger, etc.) should just be faster than the trapping rate, which is determined not only by the trap concentration, but also by the capture coefficient. All these and related effects are considered when the data is modelled by the SRH+ model developed and employed here, thus allowing the extraction of the trap concentrations without any special assumptions. Note, however, that for the bare MAPbI$_3$ sample the estimation of the trap concentration is reliable, because the influence of trap concentration alone is clearly decoupled from that of the trapping constant in the regime of trap filling and conversion from the SRH to ABC model as observed for the bare MAPbI$_3$ sample at a moderate excitation power.

Several studies have established the important role that surface defects play in determining the optoelectronic properties of perovskite thin films[58–60], yet traditional PLQY measurements do not offer a reliable method to extract the defect density in perovskite films and investigate how surface modifications influence this density of defects. Considering that a PLQY(f,P) mapping allowed us to extract the density of defects in bare MAPbI$_3$ films, we apply the same analysis to the PMMA-coated samples. We find that coating the top surface of MAPbI$_3$ with PMMA changes the picture drastically in terms of both the concentration and the nature of dominant traps. No indication of trap filling is observed in the PLQY(f,P) maps, which allows us to provide only the lower estimate for the trap density in these samples ($\approx 2 \times 10^{17}$ cm$^{-3}$ s$^{-1}$). The only part of the PLQY(f,P) map where the trap concentration and the trapping rate are decoupled is the region in which PLQY saturates, so the estimation of the high-limit of the trap concentration is not reliable due to dependence of this region on parameters related to the Auger processes. The strong increase in the trap concentration is accompanied by a decrease of the trapping rate constant $k_t$ and the nonradiative recombination rate constant $k_n$ by at least one order of magnitude. This can be interpreted by considering the traps in the PMMA-coated sample to be more prevalent, yet "weaker" than those in the bare MAPbI$_3$ sample in terms of the trapping and recombination rate introduced by each of these traps. These results suggest that the addition of PMMA at the top surface leads to the creation of weak traps, which, however, due to their very large concentration

override the effect of the stronger, yet less common, traps present in MAPbI$_3$ films that did not undergo the surface treatment.

We note that coating with polymers (including PMMA) and organic molecules in general is a common method employed in literature to protect MAPbI$_3$ samples from environmental effects when performing PL studies[61,62], and also to reduce NR recombination and improve PLQY of the material[63]. Yet our results, using PMMA as an example, reveal that such a treatment fundamentally modifies the photophysics in the perovskite layer. More importantly, the supreme sensitivity of PLQY(f,P) mapping method to the influences of interfacial modifications illustrates its efficacy for studying charge carrier dynamics not only in films, but also in multilayers and complete photovoltaic devices.

Another question that remains under debate in the perovskite community is the role of bulk defects on charge carrier dynamics in perovskite films. While some reports claim that such bulk defects, found for example at the grain boundaries, do not influence charge recombination[64,65] other reports suggest such defects influence the optoelectronic quality of the perovskite layer[66,67]. Considering these contradicting reports, it is clear that traditional PLQY measurements are not capable to idenitfy the role of bulk defects. We believe that PLQY(f,P) map is the best possible fingerprint of the sample in the context of its charge recombination pathways and may aid at resolving this and other open questions in the field. We predict that this non-invasive, simple and non-expensive method will find practical applications in controlling and optimizing semiconducting materials and the devices that are based on them.

## Conclusions

To summarize, we examined the validity of the commonly employed ABC and SRH kinetic models in describing the charge dynamics of metal halide perovskite MAPbI$_3$ semiconductor. For this purpose, we developed a novel experimental methodology based on PL measurements (PLQY and time resolved decays) performed in the two-dimensional space of the excitation energy and the repetition frequency of the laser pulses. The measured PLQY maps allow for an unmistakable distinction between samples, and more importantly, between the single-pulse and quasi-continuous excitation regimes.

We found that neither the ABC nor the SRH model can explain the complete PLQY maps for MAPbI$_3$ samples and predict the PL decays at the same time. Each model is valid only in a limited range of parameters, which may strongly vary between different samples. On the other hand, we show that the extension of the SRH model by the addition of Auger recombination and Auger trapping (SRH+ model) results in an excellent fit of the complete PLQY maps for all the studied samples. Nevertheless, even this extended model systematically underestimate the PL decay rates at high pulse fluences pointing towards the existence of additional processes in MAPbI$_3$ which must be considered to fully explain the charge carrier dynamics.

Our study clearly shows that neither PL decay nor PLQY data alone are sufficient to elucidate the photophysical processes in perovskite semiconductors. Instead, a combined PLQY mapping and time-resolved PL decays should be used to elucidate the excitation dynamics and energy loss mechanisms in luminescent semiconductors. Our experimental approach provides a strict criteria for testing any theoretic model of charge dynamics which is the requirement to be able to fit PLQY(f,P) map and PL decays at different powers and pulse repetition rates.

## Methods

**Thin film preparation**. All samples were prepared from same perovskite precursor which was prepared with 1:3 molar ratio of lead acetate trihydrate and methylammonium iodide dissolved in dimethylformamide (Supplementary Note 5). For the

samples with PMMA between the glass and perovskite layer, PMMA was spin-coated on the clean substrates at 3000 rpm for 30 s and annealed at 100 °C for 10 min. The perovskite precursor was spin-coated at 2000 rpm for 60 s on glass or glass/PMMA substrates, following by a 25 s dry air blowing, a 5 min room temperature drying and a 10 min 100 °C annealing. For the samples with PMMA on top of the perovskite layer, no further annealing was applied after the PMMA deposition.

**PL measurements**. Photoluminescence microscopy measurements were carried out using a home-built wide-field microscope based on an inverted fluorescence microscope (Olympus IX-71) (Supplementary Note 1). We used 485 nm pulsed laser (ca. 150 ps pulse duration) driven by Sepia PDL 808 controller (PicoQuant) for excitation with repetition rate tuned from 100 Hz to 80 MHz. The laser irradiated the sample through an objective lens (Olympus ×40, NA = 0.6) with ~30 μm excitation spot size. The emission of the sample was collected by the same objective and detected by an EMCCD camera (Princeton Instruments, ProEM 512B). Two motorized neutral optical density (OD) filter wheels were used: one in the excitation beam path in order to vary the excitation fluence over 4 orders of magnitude and one in the emission path to avoid saturation of the EMCCD camera. The whole measurement of a PLQY($f,P$) map was fully automated and took ~3 h (see Supplementary Note 2 for details). Automation was crucial for avoiding human errors in the measurements of so many data points (about 100 data points per PLQY($f,P$) map and to minimize light exposure of the sample.

Time-resolved photoluminescence (TRPL) measurements were carried out using the same microscope, by adding a beam splitter in front of the EMCCD and redirecting a part of the emission light to a single photon counting detector (Picoquant PMA Hybrid-42) connected to a TCSPC module (Picoharp 300).

Absolute PLQY measurements were performed using a 150 mm Spectralon Integrating Sphere (Quanta-φ, Horiba) coupled through an optical fibre to a compact spectrometer (Thorlabs CCS200). Sample PL was excited by the same laser with 80 MHz excitation repetition rate and 0.01 W/cm$^2$ excitation power density. This reference point was then used to calculate the absolute PLQY for all pulse fluences and frequency combinations (Supplementary Notes 2 and 3).

It is important to stress that the whole acquisition of PLQY($f,P$) was fully automated and the sample was exposed to light only for the measurements. This led to a rather small total irradiation dose of about 200 J/cm$^2$ (equivalent to 2000 s of 1 Sun power) per one PLQY($f,P$) map which accumulated over 85 acquisitions during about 4 h for one PLQY map. Note, that 90% of this doze was accumulated with the maximum power P5 which gives 1600 Sun when the highest frequency 80 MHz is used. This allowed us to have minimal effects of light induced PL enhancement/bleaching on the measurements.

## Data availability
The data that support the findings of this study are available from the corresponding authors upon reasonable request.

## Code availability
The codes and algorithms used for data fitting are available from the corresponding authors upon reasonable request.

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

## Acknowledgements

This work was supported by the Swedish Research Council (2016-04433) and Knut and Alice Wallenberg foundation (2016.0059). J.L. thanks China Scholarship Council (CSC No. 201608530162) for a Ph.D. scholarship. Theoretical work was supported by the Russian Science foundation Project (20-12-00202). P.A.F. and S.S. thank the Wenner-Gren foundation for the visiting (GFOh2018-0020) and postdoctoral (UPD2019-0230) scholarships. This work was supported by the European Research Council (ERC) under the European Union's Horizon 2020 research and innovation programme (ERC Grant Agreement No. 714067, ENERGYMAPS). Y.V. and Q.A. also thank the Deutsche Forschungsgemeinschaft (DFG) for funding the <PERFECT PVs> project (Grant No. 424216076) in the framework of SPP 2196. We thank Dr. Fabian Paulus for performing and analysing the XRD measurements and Prof. Jana Zaumseil for providing access to the XRD facilities.

## Author contributions

I.G.S. conceived and planned the experiments with input from P.A.F. and A.K., A.K. designed and built the automated PLQY mapping setup with contributions from I.G.S., A.K., and A.Y. performed PLQY mapping and PL decay measurements. Q.A., J.L., and S.S. prepared the samples and carried out sample characterization. Q.A. performed the UV–vis and SEM measurements. P.A.F. developed the theory, carried out the modelling and wrote the theoretical part of the manuscript. I.G.S., P.A.F. and Y.V. determined the main ideas of the study and supervised the project. I.S. wrote the manuscript with great contributions by Y.V. and P.A.F. All authors have discussed the results and commented on the final manuscript.

## Funding

## Competing interests

The authors declare no competing interests.
