## [Peer Review File · Nature Communications]

REVIEWERS' COMMENTS

Reviewer #1 (Remarks to the Author)

In my original review of this manuscript I noted that the individual elements of the work in their own right did not warrant sufficient novelty for Nature suite, but as a whole, the methodology development, applications to perovskites and model development were collectively of sufficient interest. That remains the case, and there have also been improvements to the paper in response to all reviewers comments. Responses to my comments were detailed and on the whole satisfactory - in particular the questions around diffusion, trap distributions, fluence and carrier regimes, etc. I would note these tend to be more pertinent to the case of 'very' thin film solar cells such as these perovskite based systems. Hence, I would conclude the manuscript to be improved and suitable for publication in Nature Communications.

Reviewer #2 (Remarks to the Author):

I have carefully reread the manuscript with the responses of the authors to the reviewers as reference. The manuscript is considerably improved from the previous version with stronger justification for the claims than before. I think the manuscript is now ready for publication. I still do not think there is much new insight beyond the realization that rep rate is an important parameter and much can be learned by varying it but I concede that in the perovskite world this is not well understood and as such the paper is useful to the community and is a great resource for the many groups out there doing research on this system. To that effect, I wonder if the authors have considered making some of their fitting and modeling code publicly available? It would perhaps be very useful to the community to have access to this.

Second in reading through the manuscript and thinking about the high excitation regime, perhaps something to consider is whether hot carrier effects play a role here. Many authors have found that at very high excitation energies and high pulse intensities, carriers stay hot even into the ns regime presumably because of hot-phonon bottleneck effects. This would impact recombination dynamics. At the same time, at high injection densities, bandgap renormalization starts playing a role. This would locally lower the bandgap where carrier densities are high. This would slow down diffusion from the initial spot and might locally induce extra recombination above what you expect from the carriers equilibrating throughout the film. Without doing more simulations it is hard to understand what the precise effect could be but perhaps an interesting avenue for further research. So I recommend this paper now for publication in Nat.Comm.

Small thing: Bottom of Page 16: the end of the sentence after "The latter is very important, because" is missing.

Reviewer #3 (Remarks to the Author):

The authors now made a clear case in their reply that the main selling point of the paper is the combination of the transients with the absolute intensity of the luminescence at a given fluence and repetition rate. I think, the paper is now substantially improved and could be published after minor revisions.

1) For me there is always a big difference between internal and external quantum efficiency and therefore I try to avoid the term PLQY at all costs. It just doesnt have internal or external in the name and is therefore confusing to start with. You think it should be the external quantum efficiency but you can never be sure. Too often you see equations where the PLQY is defined as the ratio of rates rather than fluxes. This difference is crucial to understand the ratio between the rates and rate constants and the measured PLQY (see <https://arxiv.org/abs/2010.12950> by some of the current authors).

The authors write the PLQY (i.e. the external luminescence quantum efficiency) as the ratio of rates, which made me convinced everything is wrong for at least an hour (wrote a review that I then had to delete) before I figured out that it is still probably ok the way they have defined their radiative

recombination coefficient. However, I would advise the authors to never start an equation with the external QE being given by the ratio of rates but always by the ratio of fluxes that may then simplify to give the ratio of rates if certain constants include information on photon recycling. In this respect I find the above paper on arxiv much more clear. The other problem with this approach is that it only works in the limit where parasitic absorption is not a thing, which is certainly a good assumption for films on glass but not for any other type of sample.

2) Trap filling. Trap filling and photodoping is something that only happens if the trap is previously empty. The authors only study acceptor like traps, therefore this only applies to traps that are above the equilibrium Fermi level so lets say roughly above midgap. Lets assume the trap is empty, then it could be filled by moving the electron Fermi level above the trap and making electron capture much more efficient than hole capture. This would then cause an increase in free holes which should counteract this whole effect so that the photodoping can only persist if the capture rate for electrons $c_n \cdot n \gg c_p \cdot p$, where c_n and c_p are rate constants for capturing electrons and holes. Given that p might $\gg n$ at some point $c_n \gg c_p$ must overcompensate that. So the difference in capture coefficients has to be huge if I understand correctly. I just want to check that the authors agree with all of the above and that this is really how it is supposed to work. Furthermore, in any situation where there is not a single shot or a steady state experiment, the occupation of the trap before the laser pulse should be different to the one in the dark at zero V and should be very important for the final result. How is that taken into account and what is the assumed initial trap occupation?

A few smaller things. TOC Figure quazi to quasi

Definition of rate constants as rates in the SI and partly also in the main paper should be corrected. Line 295: It is probably not clear to many readers why there would be a difference between the slope of n-type or p-type perovskites. The reason for the different slopes coming out of the model is the assumption of acceptor like defects. This should be mentioned in this context. If donor like defects were assumed to be dominant, it would be the other way round. But I also assume if the density of defects is small relative to the doping density, there shouldnt be a difference in slope at all. Correct?

Reviewer #4 (Remarks to the Author):

Kiligaridis and co-workers investigate the photoluminescence from lead halide perovskite thin films under pulsed laser excitation, in particular the photoluminescence quantum yield as a function of the laser repetition rate. They use the results to identify two regimes, depending on whether long-lived trapped optical excitations survive or not from one pulse to the next.

Although nor the technique, nor the samples are particularly new, the issue of traps detrimental to photoconversion is still very actual and often addressed with time-resolved photoluminescence studies. Therefore, it seems to me that the topic may be of interest for the community studying the optical properties of perovskites for solar cells and LEDs.

Coming to the scientific part, it seems to me that the analysis proposed by the authors helps separating quantitatively the Markov dynamics, that can be described by rate equations (the ABC model), from non-Markovian decays that instead depend on the previous history of optical excitations and therefore cannot be described with a single density, but require more parameters, like the direct inclusion of trap filling, or even worse may include spatial diffusion, either lateral, vertical or both.

The manuscript would improve if the authors were able to sort out spatial diffusion effects from traps. Spatial diffusion depends on the specific excitation they employ and may be reduced e.g. by using a laser wavelength closer to the bandgap to achieve a more homogeneous excitation profile in the vertical direction, and by increasing the excitation spot size to reduce lateral inhomogeneity. On the other hand, traps are a property of the sample, and as such much more interesting. Also in this case, illustrating how the technique provide some feedback on the materials quality may be interesting, by varying the sample thickness to discriminate surface and bulk traps, or even better by applying treatments to the sample affecting and curing trap states.

Apart from such points, I recommend the manuscript as suitable for publication in Nature Communications.

----- Response letter -----

We are grateful to all reviewers for their time, comments, and very positive response on our manuscript. Please, see below our detail reply to the comments. The changes in the revised version of the manuscript are highlighted by blue font colour.

Reviewer #1

Reviewer #1, Comment #1:

In my original review of this manuscript I noted that the individual elements of the work in their own right did not warrant sufficient novelty for Nature suite, but as a whole, the methodology development, applications to perovskites and model development were collectively of sufficient interest. That remains the case, and there have also been improvements to the paper in response to all reviewers comments. Responses to my comments were detailed and on the whole satisfactory - in particular the questions around diffusion, trap distributions, fluence and carrier regimes, etc. I would note these tend to be more pertinent to the case of 'very' thin film solar cells such as these perovskite based systems. Hence, I would conclude the manuscript to be improved and suitable for publication in Nature Communications.

Response 1-1.

We are grateful to the reviewer for so positive evaluation of our work.

Reviewer #2

Reviewer #2, Comment #1:

I have carefully reread the manuscript with the responses of the authors to the reviewers as reference. The manuscript is considerably improved from the previous version with stronger justification for the claims than before. I think the manuscript is now ready for publication. I still do not think there is much new insight beyond the realization that rep rate is an important parameter and much can be learned by varying it but I concede that in the perovskite world this is not well understood and as such the paper is useful to the community and is a great resource for the many groups out there doing research on this system.

Response 2-1.

We are grateful to the reviewer for the positive evaluation of our work for publication in Nature Communication.

Reviewer #2, Comment #2:

To that effect, I wonder if the authors have considered making some of their fitting and modeling code publicly available? It would perhaps be very useful to the community to have access to this.

Response 2-2.

Yes, we are planning to make the codes and procedures available upon reasonable request. In the future, since we continue working on several more projects using the same techniques and fitting, we will try to make the codes suitable for publication on a code repository site like github.

Reviewer #2, Comment #3:

Second in reading through the manuscript and thinking about the high excitation regime, perhaps something to consider is whether hot carrier effects play a role here. Many authors have found that at very high excitation energies and high pulse intensities, carriers stay hot even into the ns regime presumably because of hot-phonon bottleneck effects. This would impact recombination dynamics. At the same time, at high injection densities, bandgap renormalization starts playing a role. This would locally lower the bandgap where carrier densities are high. This would slow down diffusion from the initial spot and might locally induce extra recombination above what you expect from the carriers equilibrating throughout the film. Without doing more simulations it is hard to understand what the precise effect could be but perhaps an interesting avenue for further research.

Response 2-3.

We thank that reviewer for this suggestion, indeed, at high concentration regime these effects need to be seriously considered as we plan to do in our future research.

Reviewer #2, Comment #4:

Small thing: Bottom of Page 16: the end of the sentence after "The latter is very important, because" is missing.

Response 2-4.

The broken sentence was deleted, thank you very much.

Reviewer #3**Reviewer #3, Comment #1**

The authors now made a clear case in their reply that the main selling point of the paper is the combination of the transients with the absolute intensity of the luminescence at a given fluence and repetition rate. I think, the paper is now substantially improved and could be published after minor revisions.

Response 3-1.

We thank the reviewer for the positive evaluation of our work.

Reviewer #3, Comment #2

1) For me there is always a big difference between internal and external quantum efficiency and therefore I try to avoid the term PLQY at all costs. It just doesn't have internal or external in the name and is therefore confusing to start with. You think it should be the external quantum efficiency but you can never be sure. Too often you see equations where the PLQY is defined as the ratio of rates rather than fluxes. This difference is crucial to understand the ratio between the rates and rate constants and the measured PLQY (see <https://arxiv.org/abs/2010.12950> by some of the current authors).

The authors write the PLQY (i.e. the external luminescence quantum efficiency) as the ratio of rates, which made me convinced everything is wrong for at least an hour (wrote a review that I then had to delete) before I figured out that it is still probably ok the way they have defined their radiative recombination coefficient. However, I would advise the authors to never start an equation with the external QE being given by the ratio of rates but always by the ratio of fluxes that may then simplify to give the ratio of rates if certain constants include information on photon recycling. In this respect I find

the above paper on arxiv much more clear. The other problem with this approach is that it only works in the limit where parasitic absorption is not a thing, which is certainly a good assumption for films on glass but not for any other type of sample.

Response 3-2.

We fully agree with the reviewer. The equation (9.4) in Supplementary Note 9 gives the external light emission flux (number of photons emitted per second from one cubic centimetre of the material) for the SRH+ model:

$$PL(t) = k_r n(t) p(t)$$

As discussed in the Supplementary Note 10, k_r is the renormalized radiative recombination rate. The effects of the photon reabsorption are already taken into account in its value.

The absorbed light flux is denoted by $G(t)$ (number of absorbed photons per second within one cubic centimeter) in Eqs. (9.1-9.3) in Supplementary Note 9. In the pulse experiment, we introduced the averaged flux G , defined as the integral number of the generated carriers in the cubic centimetre per pulse divided by the pulse period T .

$$G = \frac{n_0}{T}$$

where

$$n_0 = \int_0^T G(t) dt$$

The external PLQY is defined as the ratio between the averaged PL flux and G :

$$\overline{PL} = \frac{1}{T} \int_0^T k_r n(t) p(t) dt$$

$$PLQY = \frac{\overline{PL}}{G} = \frac{1}{TG} \int_0^T k_r n(t) p(t) dt = \frac{1}{n_0} \int_0^T k_r n(t) p(t) dt$$

This expression is given Eq. (9.22) in Supplementary Note 9. Analogous expression for the ABC model is given by Eq. (9.8).

Thus, our definition of the PL quantum yield is the ratio between the averaged flux of the external light emission and the averaged flux of the absorbed light. It is equivalent to the definition of the external QE given by the reviewer.

We agree with the reviewer that in our samples (perovskite on glass) so-called parasitic absorption (e.g. absorption by e.g. glass or anything else than perovskite itself) is negligible and that is why it is not considered. This is mentioned now explicitly in the Supplementary Note 10.

In the main text we explicitly state now that we are dealing with the external PLQY (page 3). Also, in the Supplementary Note 9 we rephrased the first paragraph which now reads as:

“PLQY calculated in this section is the so-called external PLQY because we operate with the experimentally determined PL and excitation intensities. Photon recycling is included indirectly in the theoretical models, because the rate constants of radiative recombination and Auger trapping can be seen as re-normalized constants corresponding to the light/energy propagation conditions for the particular sample, see Supplementary Note 10 for details”.

Following the suggestion by the reviewer, we also added in the main text terms emission and absorption photon flux in the equations for PLQY. From the text it is clear now that that the rate constants we are dealing with are renormalized constants taking into account processes of photon reabsorption in the perovskite materials.

Reviewer #3, Comment #3

2) Trap filling. Trap filling and photodoping is something that only happens if the trap is previously empty. The authors only study acceptor like traps, therefore this only applies to traps that are above the equilibrium Fermi level so lets say roughly above midgap. Lets assume the trap is empty, then it could be filled by moving the electron Fermi level above the trap and making electron capture much more efficient than hole capture. This would then cause an increase in free holes which should counteract this whole effect so that the photodoping can only persist if the capture rate for electrons $c_n \cdot n \gg c_p \cdot p$, where c_n and c_p are rate constants for capturing electrons and holes. Given that p might $\gg n$ at some point $c_n \gg c_p$ must overcompensate that. So the difference in capture coefficients has to be huge if I understand correctly. I just want to check that the authors agree with all of the above and that this is really how it is supposed to work.

Response 3-3.

We agree with this picture described by the reviewer. The trap states in the SRH+ model are placed above the Fermi level and unpopulated in the absence of light excitation. Under light excitation, the generated electrons in the conduction band can be trapped. The flux of electrons' trapping is proportional to their density n as well as to the density of empty traps $(N - n_t)$. In the SRH+ model this flux has the following form:

$$k_t(N - n_t)n$$

An electron in the trap state can be considered as a hole trap. The holes' trapping flux has to be proportional to the hole density p and the density of the empty hole traps n_t . This flux is presented in SRH+ model by the following term:

$$k_n n_t p$$

When the system reaches the trap filling regime, the density of trapped electrons is limited by trap density N . At the further increasing of the excitation intensity the density of electrons and holes became much larger than N and, since

$$n \approx p$$

the condition noted by the Reviewer can be written as

$$k_t(N - n_t)n \gg k_n n_t n$$

from which it follows:

$$k_t \gg k_n$$

This condition is the part of the trap filling regime condition reported in our manuscript (see p. 29 in the Supplementary Note):

$$k_t \gg k_n, k_r$$

Please, see Supplementary Note 9 for more details.

Reviewer #3, Comment #4

Furthermore, in any situation where

there is not a single shot or a steady state experiment, the occupation of the trap before the laser pulse should be different to the one in the dark at zero V and should be very important for the final result. How is that taken into account and what is the assumed initial trap occupation?

Response 3-4.

The Reviewer is absolutely correct. The trap density before the laser pulse as well as that of electrons and holes are very important for the PLQY calculations within the SHR+ model. After applying many pulses, the carrier densities reach so-called periodic solution which satisfy the following periodic conditions:

$$n(0) = n(T) + n_0$$

$$p(0) = p(T) + n_0$$

$$n_t(0) = n_t(T)$$

The periodicity here means that n , p and n_t values right before a laser pulse are exactly the same as before the next laser pulse. Our algorithm includes finding the periodic solution of Eqs. (9.18-9.20) as the first part of the of PLQY calculation for the pulse experiment.

Reviewer #3, Comment #5

A few smaller things. TOC Figure quazi to quasi.

Response 3-5.

The mistake was corrected, thank you.

Reviewer #3, Comment #6

Definition of rate constants as rates in the SI and partly also in the main paper should be corrected.

Response 3-6.

We thank the reviewer for noticing this inconsistency in the language. We have checked the whole manuscript to made sure that all coefficients (k_r , k_n , k_a ...) are called "rate constants", while the speeds of the corresponding processes are called "rates".

Reviewer #3, Comment #7

Line 295: It is probably not clear to many readers why there would be a difference between the slope of n-type or p-type perovskites. The reason for the different slopes coming out of the model is the assumption of acceptor like defects. This should be mentioned in this context. If donor like defects were assumed to be dominant, it would be the other way round. But I also assume if the density of defects is small relative to the doping density, there shouldnt be a difference in slope at all. Correct?

Response 3-7.

Yes, the reviewer is absolutely correct, we are grateful for this note. The situation is symmetrical and dependent on the type of the trap (electron and hole trap). We clarified it in the revised version by adding a few words in page 8 (top).

Reviewer #4

Reviewer #4, Comment #1

Kiligaridis and co-workers investigate the photoluminescence from lead halide perovskite thin films under pulsed laser excitation, in particular the photoluminescence quantum yield as a function of the laser repetition rate. They use the results to identify two regimes, depending on whether long-lived trapped optical excitations survive or not from one pulse to the next. Although nor the technique, nor the samples are particularly new, the issue of traps detrimental to photoconversion is still very actual and often addressed with time-resolved photoluminescence studies. Therefore, it seems to me that the topic may be of interest for the community studying the optical properties of perovskites for solar cells and LEDs.

Coming to the scientific part, it seems to me that the analysis proposed by the authors helps separating quantitatively the Markov dynamics, that can be described by rate equations (the ABC model), from non-Markovian decays that instead depend on the previous history of optical excitations and therefore cannot be described with a single density, but require more parameters, like the direct inclusion of trap filling, or even worse may include spatial diffusion, either lateral, vertical or both.

The manuscript would improve if the authors were able to sort out spatial diffusion effects from traps. Spatial diffusion depends on the specific excitation they employ and may be reduced e.g. by using a laser wavelength closer to the bandgap to achieve a more homogeneous excitation profile in the vertical direction, and by increasing the excitation spot size to reduce lateral inhomogeneity. On the other hand, traps are a property of the sample, and as such much more interesting. Also in this case, illustrating how the technique provide some feedback on the materials quality may be interesting, by varying the sample thickness to discriminate surface and bulk traps, or even better by applying treatments to the sample affecting and curing trap states. Apart from such points, I recommend the manuscript as suitable for publication in Nature Communications.

Response 4-1.

We thank the reviewer for the positive assessment of our work.

The reviewer proposes several experiments that we indeed considered in detail. The idea of changing the thickness of the perovskite layer in order to discriminate surface and bulk traps was something we

examined closely. The main issue we encountered is that fabrication of thinner perovskite layers leads to dramatic change in their microstructure, i.e. a significant reduction in the average grain size, which will alter the defect concentration of the film simply due to the presence of grain boundaries. Consequently, both the surface and bulk trap densities change when making the film thinner, so it would not be possible to distinguish between them by changing the film thickness. Please, see Figure S1 in <https://doi.org/10.1016/j.matt.2021.02.020> (An et al., Matter 4 (2021) 1–19).

Regarding the lateral inhomogeneity, the excitation spot we use is relatively large at dozens of micrometres, which means that lateral diffusion would not play a role. We also considered changing the excitation wavelength, however this introduces additional effects apart from changing the probing depth, such as the different excess energy provided to the sample for excitation above the bandgap.

To summarise, we thank the referee for the suggestions, and we definitely consider the study of diffusion to be critically important, however we feel that it falls beyond the scope of this manuscript and will be subject to future investigations.